# DOM-Q-NET:
# GROUNDED RL ON STRUCTURED LANGUAGE

**Sheng Jia**
University of Toronto
Vector Institute
sheng.jia@utoronto.ca

**Jamie Kiros**
Google Brain
kiros@google.com

**Jimmy Ba**
University of Toronto
Vector Institute
jba@cs.toronto.ca

## ABSTRACT

Building agents to interact with the web would allow for significant improvements in knowledge understanding and representation learning. However, web navigation tasks are difficult for current deep reinforcement learning (RL) models due to the large discrete action space and the varying number of actions between the states. In this work, we introduce DOM-Q-NET, a novel architecture for RL-based web navigation to address both of these problems. It parametrizes Q functions with separate networks for different action categories: clicking a DOM element and typing a string input. Our model utilizes a graph neural network to represent the tree-structured HTML of a standard web page. We demonstrate the capabilities of our model on the MiniWoB environment where we can match or outperform existing work without the use of expert demonstrations. Furthermore, we show 2x improvements in sample efficiency when training in the multi-task setting, allowing our model to transfer learned behaviours across tasks.

## 1 INTRODUCTION

Over the past years, deep reinforcement learning (RL) has shown a huge success in solving tasks such as playing arcade games (Mnih et al., 2015) and manipulating robotic arms (Levine et al., 2016). Recent advances in neural networks allow RL agents to learn control policies from raw pixels without feature engineering by human experts. However, most of the deep RL methods focus on solving problems in either simulated physics environments where the inputs to the agents are joint angles and velocities, or simulated video games where the inputs are rendered graphics. Agents trained in such simulated environments have little knowledge about the rich semantics of the world.

The World Wide Web (WWW) is a rich repository of knowledge about the real world. To navigate in this complex web environment, an agent needs to learn about the semantic meaning of texts, images and the relationships between them. Each action corresponds to interacting with the Document Object Model (DOM) from tree-structured HTML. Tasks like finding a friend on a social network, clicking an interesting link, and rating a place on Google Maps can be framed as accessing a particular DOM element and modifying its value with the user input.

In contrast to Atari games, the difficulty of web tasks comes from their diversity, large action space, and sparse reward signals. A common solution for the agent is to mimic the expert demonstration by imitation learning in the previous works (Shi et al., 2017; Liu et al., 2018). Liu et al. (2018) achieved state-of-the-art performance with very few expert demonstrations in the MiniWoB (Shi et al., 2017) benchmark tasks, but their exploration policy requires constrained action sets, hand-crafted with expert knowledge in HTML.

In this work, our contribution is to propose a novel architecture, DOM-Q-NET, that parametrizes factorized Q functions for web navigation, which can be trained to match or outperform existing work on MiniWoB without using any expert demonstration. Graph Neural Network (Scarselli et al., 2009; Li et al., 2016; Kipf & Welling, 2016) is used as the main backbone to provide three levels of state and action representations.

In particular, our model uses the neural message passing and the readout (Gilmer et al., 2017) of the *local* DOM representations to produce *neighbor* and *global* representations for the web page.

We also propose to use three separate multilayer perceptrons (MLP) (Rumelhart et al., 1985) to parametrize a factorized Q function for different action categories: "click", "type" and "mode". The entire architecture is fully differentiable, and all of its components are jointly trained.

Moreover, we evaluate our model on multitask learning of web navigation tasks, and demonstrate the transferability of learned behaviors on the web interface. To our knowledge, this is the first instance that an RL agent solves multiple tasks in the MiniWoB at once. We show that the multi-task agent achieves an average of 2x sample efficiency comparing to the single-task agent.

## 2 BACKGROUND

### 2.1 REPRESENTING WEB PAGES USING DOMS

The Document Object Model (DOM) is a programming interface for HTML documents and it defines the logical structure of such documents. DOMs are connected in a tree structure, and we frame web navigation as accessing a DOM and optionally modifying it by the user input. As an elementary object, each DOM has a "tag" and other attributes such as "class", "is focused", similar to the *object* in Object Oriented Programming. Browsers use those attributes to render web pages for users.

### 2.2 REINFORCEMENT LEARNING

In the traditional reinforcement learning setting, an agent interacts with an infinite-horizon, discounted Markov Decision Process (MDP) to maximize its total discounted future rewards. An MDP is defined as a tuple $(\mathcal{S}, \mathcal{A}, T, R, \gamma)$ where $\mathcal{S}$ and $\mathcal{A}$ are the state space and the action space respectively, $T(s'|s, a)$ is the transition probability of reaching state $s' \in \mathcal{S}$ by taking action $a \in \mathcal{A}$ from $s \in \mathcal{S}$, $R$ is the immediate reward by the transition, and $\gamma$ is a discount factor. The Q-value function for a tuple of actions is defined to be $Q^\pi(s, a) = \mathbb{E}[\sum_{t=0}^{T} \gamma^t r_t | s_0 = s, a_0 = a]$, where T is the number of timesteps till termination. The formula represents the expected future discounted reward starting from state $s$, performing action $a$ and following the policy until termination. The optimal Q-value function $Q^*(s, a) = max_\pi Q^\pi(s, a), \forall s \in \mathcal{S}, a \in \mathcal{A}$ (Sutton & Barto, 1998) satisfies the Bellman optimality equation $Q^*(s, a) = \mathbb{E}_{s'}[r + \gamma \max_{a' \in \mathcal{A}} Q^*(s', a')]$.

### 2.3 GRAPH NEURAL NETWORKS

For an undirected graph $G = (V, E)$, the Message Passing Neural Network (MPNN) framework (Gilmer et al., 2017) formulates two phases of the forward pass to update the node-level feature representations $h_v$, where $v \in V$, and graph-level feature vector $\hat{y}$. The message passing phase updates hidden states of each node by applying a vertex update function $U_t$ over the current hidden state and the message, $h_v^{t+1} = U_t(h_v^t, m_v^{t+1})$, where the passed message $m_v^{t+1}$ is computed as $m_v^{t+1} = \sum_{\omega \in N(v)} M_t(h_v^t, h_w^t, e_{vw})$. $N(v)$ denotes the neighbors of $v$ in $G$, and $e_{vw}$ is an edge feature. This process runs for T timesteps. The readout phase uses the readout function R, and computes the graph-level feature vector $\hat{y} = R(h_v^T | v \in G)$.

### 2.4 REINFORCEMENT LEARNING WITH GRAPH NEURAL NETWORKS

There has been work in robot locomotion that uses graph neural networks (GNNs) to model the physical body (Wang et al., 2018; Hamrick et al., 2018). NerveNet demonstrates that policies learned with GNN transfers better to other learning tasks than policies learned with MLP (Wang et al., 2018). It uses GNNs to parametrize the entire policy whereas DOM-Q-NET uses GNNs to provide representational modules for factorized Q functions. Note that the graph structure of a robot is static whereas the graph structure of a web page can change at each time step. Locomotion-based control tasks provide dense rewards whereas web navigation tasks are sparse reward problems with only 0/1 reward at the end of the episode. For our tasks, the model also needs to account for the dependency of actions on goal instructions.

## 2.5 PREVIOUS WORK ON RL ON WEB INTERFACES

Shi et al. (2017) constructed benchmark tasks, Mini World of Bits (MiniWoB), that consist of many toy tasks of web navigation. This environment provides both the image and HTML of a web page. Their work showed that the agent using the visual input cannot solve most of the tasks, even given the demonstrations. Then Liu et al. (2018) proposed DOM-NET architecture that uses a series of attention between DOM elements and the goal. With their workflow guided-exploration that uses the formal language to constrain the action space of an agent, they achieved state-of-the-art performance and sample efficiency in using demonstrations. Unlike these previous approaches, we aim to tackle web navigation without any expert demonstration or prior knowledge.

## 3 NEURAL DOM Q NETWORK

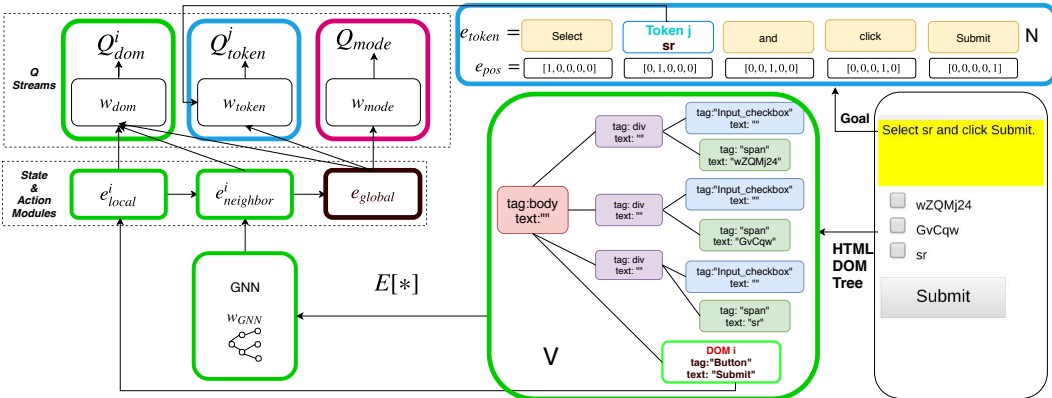

Figure 1: Given the web page on the right, its DOM tree representation is shown as a graph where each DOM represents a node from $V$. Different colors indicate different tag attributes of DOMs. DOMs are embedded as a local module, $\boldsymbol{e}_{local}$, and propagated by a GNN to produce a neighbor module, $\boldsymbol{e}_{neighbor}$. The global module, $\boldsymbol{e}_{global}$, is aggregated from the neighbor module. The $Q_{dom}$ stream uses all three modules whereas $Q_{token}$ and $Q_{mode}$ streams only use the global module. Here, Q values of the 'submit' and 'sr' token are computed by $Q_{dom}$ and $Q_{token}$ respectively.

Consider the problem of navigating through multiple web pages or menus to locate a piece of information. Let $V$ be the set of DOMs in the current web page. There are often multiple goals that can be achieved in the same web environment. We consider goals that are presented to the agent in the form of a natural language sentence, e.g. "Select sr and click Submit" in Figure 1 and "Use the textbox to enter Kanesha and press Search, then find and click the 9th search result" in Figure 2. Let $\mathcal{G}$ represent the set of word tokens in the given goal sentence. The RL agent will only receive a reward if it successfully accomplishes the goal, so it is a sparse reward problem. The primary means of navigation are through interaction with the buttons and the text fields on the web pages.

There are two major challenges in representing the state-action value function for web navigation: the action space is enormous, and the number of actions can vary drastically between the states. We propose DOM-Q-NET to address both of the problems in the following.

## 3.1 ACTION SPACE FOR WEB NAVIGATION

In contrast to typical RL tasks that require choosing only one action $a$ from an action space, $\mathcal{A}$, such as choosing one from all combinations of controller's joint movements for Atari (Mnih et al., 2015), we frame acting on the web with three distinct categories of actions:

- DOM selection $a_{dom}$ chooses a single DOM in the current web page, $a_{dom} \in V$. The DOM selection covers the typical interactive actions such as clicking buttons or checkboxes as well as choosing which text box to fill in the string input.

- Word token selection $a_{token} \in \mathcal{G}$ picks a work token from the given goal sentence to fill in the selected text box. The assumption that typed string comes from the goal instruction aligns with the previous work (Liu et al., 2018).

- Mode $a_{mode} \in \{\text{click}, \text{type}\}$ tells the environment whether the agent's intention is to "click" or "type" when acting in the web page. $a_{mode}$ is represented as a binary action.

At each time step, the environment receives a tuple of actions, namely $a = (a_{dom}, a_{token}, a_{mode})$, though it does not process $a_{token}$ unless $a_{mode} = type$.

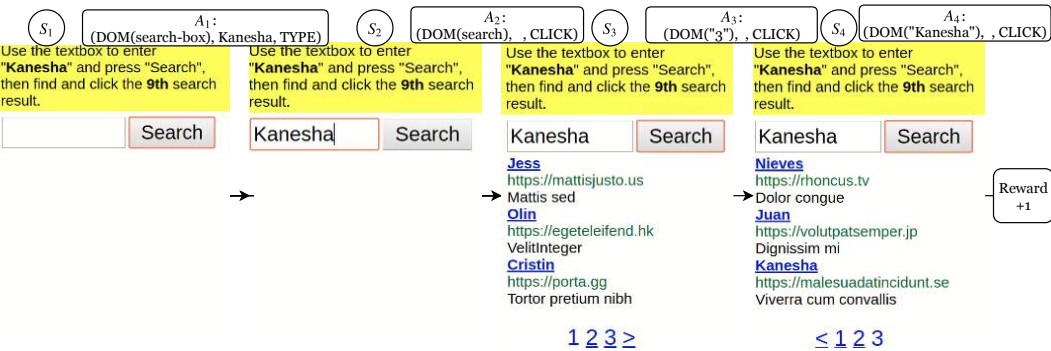

Figure 2: A successful trajectory executed by our model for *search-engine*. $S_i$ is the state, and $A_i = (a_{dom}, a_{token}, a_{mode})$ is a tuple of actions for the three distinct categories of actions at timestep i. DOM($x$) represents the index of the corresponding element $x$ in the web page.

### 3.2 FACTORIZED Q FUNCTION

One way to represent the state-action value function is to consider all the permutations of $a_{dom}$ and $a_{token}$. For example, Mnih et al. (2015) considers the permutations of joystick direction and button clicking for Atari. For MiniWoB, this introduces an enormous action space with size $|V| \times |\mathcal{G}|$. The number of DOMs and goal tokens, $|V|$ and $|\mathcal{G}|$, can reach up to 60 and 18, and the total number of actions become over $1,000$ for some hard tasks.

To reduce the action space, we consider a factorized state-action value function where the action values of $a_{dom}$ and $a_{token}$ are **independent** to each other. Formally, we define the optimal Q-value function as the sum of the individual value functions of the three action categories:

$$Q^*(s,a) = Q^*(s, a_{dom}, a_{token}, a_{mode}) = Q^*(s, a_{dom}) + Q^*(s, a_{token}) + Q^*(s, a_{mode}). \quad (1)$$

Under the independence assumption, we can find the optimal policy by selecting the greedy actions w.r.t. each Q-value function individually. Therefore, the computation cost for the optimal action of the factorized Q function is linear in the number of DOM elements and the number of word tokens rather than quadratic.

$$a^* = \left( \arg\max_{a_{dom}} Q^*(s, a_{dom}), \arg\max_{a_{token}} Q^*(s, a_{token}), \arg\max_{a_{mode}} Q^*(s, a_{mode}) \right) \quad (2)$$

### 3.3 LEARNING STATE-ACTION EMBEDDINGS OF WEB PAGES

Many actions on the web, clicking different checkboxes and filling unseen type of forms, share similar tag or class attributes. Our goal is to design a neural network architecture that effectively captures such invariance for web pages, and yet is flexible to deal with the varying number of DOM elements and goal tokens at different time steps. Furthermore, when locating a piece of information on the web, an agent needs to be aware of both the local information, e.g. the name of button and its surrounding texts, and the global information, e.g. the general theme, of the web page. The cue for clicking a particular button from the menu is likely scattered.

To address the above problem, we propose a GNN-based RL agent that computes the factorized Q-value for each DOM in the current web page, called DOM-Q-NET as shown in Figure 1. It uses additional information of tree-structured HTML to guide the learning of state-action representations, embeddings $e$, which is shared among factorized Q networks. Explicitly modeling the HTML tree structure provides the relational information among the DOM elements to the agent. Given a web page, our model learns a concatenated embedding vector $e^i = \left[ e^i_{local}, e^i_{neighbor}, e_{global} \right]$ using the low-level and high-level modules that correspond to node-level and graph-level outputs of the GNN.

**Local Module** $e^i_{local}$ is the concatenation of each embedded attribute $e_{Attr}$ of the DOM $v^i$, which includes the tag, class, focus, tampered, and text information of the DOM element. In particular, we use the maximum of cosine distance between the text and each goal token to measure the soft alignment of the DOM $v^i$ with the $j^{th}$ word embedding, $e^j_{goal}$, in the goal sentence. Liu et al. (2018) uses the exact alignment to obtain tokens that appear in the goal sentence, but our method can detect synonyms that are not exactly matched.

$$e^i_{local} = \left[ e^i_{Attr}, \max_j \left( cos(e^i_{Attr}, e^j_{goal}) \right) \right] \tag{3}$$

This provides the unpropagated action representation of clicking each DOM, and is the *skip connection* of GNNs.

**Neighbor Module** $e^i_{neighbor}$ is the node representation that incorporates the neighbor context of the DOM $v^i$ using a graph neural network. The model performs the message passing between the nodes of the tree with the weights $w_{GNN}$. The local module is used to initialize this process. $m^t$ is an intermediate state for each step of the message passing, and we adopt Gated Recurrent Units (Cho et al., 2014) for the nonlinear vertex update (Li et al., 2016). This process is performed for T number of steps to obtain the final neighbor embeddings.

$$m^{i,t+1}_{neighbor} = \sum_{k \in N(i)} w_{GNN} e^{k,t}_{neighbor}, \quad e^{i,0}_{neighbor} = e^i_{local}, \tag{4}$$

$$e^{i,t+1}_{neighbor} = GRU(e^{i,t}_{neighbor}, m^{i,t+1}_{neighbor}), \quad e^i_{neighbor} = e^{i,T}_{neighbor} \tag{5}$$

By incorporating the context information, this module contains the state representation of the current page, and the propagated action representation of clicking the particular DOM, so the Q-value function can be approximated using only this module.

**Global Module** $e_{global}$ is the high-level feature representation of the entire web page after the readout phase. It is used by all three factorized Q networks. We investigate two readout functions to obtain such global embedding with and without explicitly incorporating the goal information.

1) We use *max-pooling* to aggregate all of the DOM embeddings of the web page.

$$e_{global} = \text{maxpool} \left( \left\{ \left[ e^i_{local}, e^i_{neighbor} \right] | v^i \in V \right\} \right) \tag{6}$$

2) We use *goal-attention* with the goal vector as an attention query. This is in contrast to Velickovic et al. (2018) where the attention is used in the message passing phase, and the query is not a task dependent representation. To have the goal vector $h_{goal}$, each goal token $e_{token}$ is concatenated with the one-hot positional encoding vector $e_{pos}$, as shown in Figure 1. Next, the position-wise feed-forward network with ReLU activation is applied to each concatenated vector before max-pooling the goal representation. Motivated by Vaswani et al. (2017), we use scaled dot product attention with local embeddings as keys, and neighbor embeddings as values. Note that $E_{local}$ and $E_{neighbor}$ are packed representations of $(e^1_{local}, ..., e^V_{local})$ and $(e^1_{neighbor}, ..., e^V_{neighbor})$ respectively, where $E_{local} \in \mathbb{R}^{(V,d_k)}$, $E_{neighbor} \in \mathbb{R}^{(V,d_k)}$, and $d_k$ is the dimension of text token embeddings.

$$e_{attn} = \text{softmax}(\frac{h_{goal} E^T_{local}}{\sqrt{d_k}}) E_{neighbor}, \quad e_{global\_attn} = [e_{global}, e_{attn}] \tag{7}$$

The illustrative diagram is shown in Appendix 6.2, and a simpler method of concatenating the node-level feature with the goal vector is shown in Appendix 6.3. This method is also found to be effective in incorporating the goal information, but the size of the model increases.

**Learning** The Q-value function of choosing the DOM is parametrized by a two-layer MLP, $Q^i_{dom} = MLP(e^i; w_{dom})$, where it takes the concatenation of DOM embeddings $e^i = \left[ e^i_{local}, e^i_{neighbor}, e_{global} \right]$ as the input. Similarly, the Q-value functions for choosing the word token and the mode are computed using $MLP(e_{token}, e_{global}; w_{token})$ and $MLP(e_{global}; w_{mode})$ respectively. See Figure 1. All the model parameters including the embedding matrices are learned from scratch. Let $\theta = (E, w_{GNN}, w_{dom}, w_{token}, w_{mode})$ be the model parameters including the embedding matrices, the weights of a graph neural network, and weights of the factorized Q-value function. The model parameters are updated by minimizing the squared TD error (Sutton, 1988):

$$\min_\theta \mathbb{E}_{(s,a,r,s') \sim \text{replay}} \left[ \left( y^{DQN} - Q(s, a_{dom}; \theta) - Q(s, a_{token}; \theta) - Q(s, a_{mode}; \theta) \right)^2 \right], \quad (8)$$

where the transition pairs $(s, a, r, s')$ are sampled from the replay buffer and $y^{DQN}$ is the factorized target Q-value with the target network parameters $\theta^-$ as in the standard DQN algorithm.

$$y^{DQN} = r + \gamma \left( \max_{a'_{dom}} Q(s', a'_{dom}; \theta^-) + \max_{a'_{token}} Q(s', a'_{token}; \theta^-) + \max_{a'_{mode}} Q(s', a'_{mode}; \theta^-) \right) \quad (9)$$

### 3.4 Multitask Learning for Transferring Learned Behaviours

To assess the effectiveness of transferring learned behaviours and solving multiple tasks by our model, we train a single agent acting in multiple environments. Transitions from different tasks are collected in a shared replay buffer, and the network is updated after performing an action in each environment. See Alg.1 for details.

## 4 Experiments

We first evaluate the generalization capability of the proposed model for large action space by comparing it against previous works. Tasks with various difficulties, as defined in Appendix 6.4, are chosen from MiniWoB. Next, we investigate the gain in sample efficiency with our model from multitask learning. We perform an ablation study to justify the effectiveness of each representational module, followed by the comparisons of gains in sample efficiency from goal-attention in multitask and single task settings. Hyperparameters are explained in Appendix 6.1.

### 4.1 DOM-Q-NET Benchmark MiniWoB

We use the Q-learning algorithm, with four components of Rainbow (Hessel et al., 2018), to train our agent because web navigation tasks are sparse reward problems, and an off-policy learning with a replay buffer is more sample-efficient. The four components are DDQN (Van Hasselt et al., 2016), Prioritized replay (Schaul et al., 2016), Multi-step learning (Sutton, 1988) , and NoisyNet (Fortunato et al., 2018). To align with the settings used by Liu et al. (2018), we consider the tasks that only require clicking DOM elements and typing strings. The agent receives +1 reward if the task is completed correctly, and 0 reward otherwise. We perform $T = 3$ steps of *neural message passing* for all the tasks except *social-media*, for which we use $T = 7$ steps to address the large DOM space.

**Evaluation metric:** We plot the moving average of rewards for the last 100 episodes during training. We follow previous works (Shi et al., 2017; Liu et al., 2018), and report the success rate, which is the percentage of test episodes ending up with the reward +1. Each reported success rate is based on the average of 4 different runs, and Appendix 6.6 explains our experiment protocol.

**Results:** Figure 3 shows that DOM-Q-NET reaches 100% success rate for most of the tasks selected by Liu et al. (2018), except for *click-widget*, *social-media*, and *email-inbox*. Our model still reaches 86% success rate for *social-media*, and the use of goal-attention enables the model to solve *click-widget* and *social-media* with 100% success rate. We did not use any prior knowledge such as providing constraints on the action set during exploration, using pre-defined fields of the goal and showing expert demonstrations. Specifically, our model solves a long-horizon task, *choose-date*, that previous works with demonstrations were unable to solve. This task expects many similar actions, but has a large action space. Even using imitation learning or guided exploration, the neural network needs to learn a representation that generalizes for unseen diverse DOM states and actions, which our model proves to do.

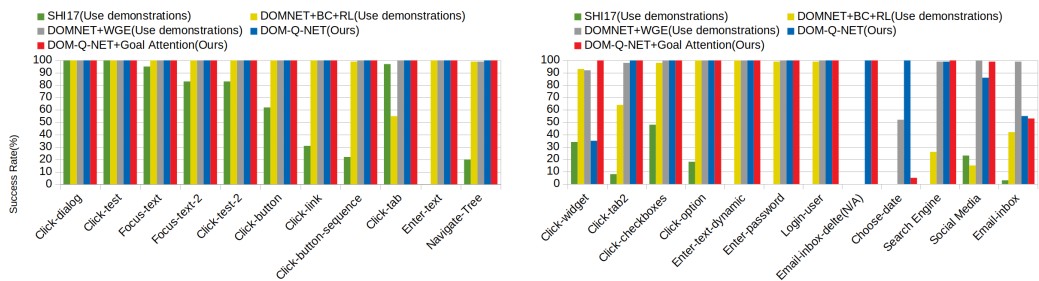

Figure 3: Performance comparisons of DOM-Q-NET with Shi et al. (2017); Liu et al. (2018)

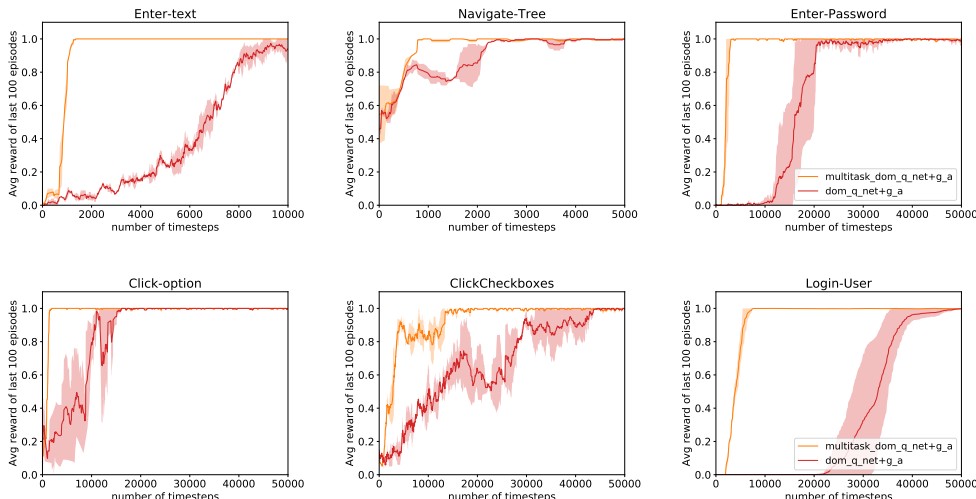

Figure 4: Multitask Comparisons: 9-multitask DOM-Q-NET with goal-attention consistently has better sample efficiency. Results for other tasks are shown in Appendix 6.7.1. g_a=goal-attention.

## 4.2 MULTITASK

Two metrics are used for comparing the sample efficiency of multitask and single-task agents.

- $M_{total}$ multitask agent: total number of frames observed upon solving all the tasks.
  $M_{total}$ single-task agents: sum of the number of frames observed for solving each task.

- $M_{task}$: number of frames observed for solving a specific task.

We trained a multitask agent solving 9 tasks with 2x sample efficiency, using about $M_{total} = 63000$ frames, whereas the single-task agents use $M_{total} = 127000$ frames combined. Figure 4 shows the plots for 6 out of the 9 tasks. In particular, *login-user* and *click-checkboxes* are solved with 40000 fewer frames using multitask learning, but such gains are not as obvious when the task is simple, as in the case of *navigate-tree*. Next we included two hard tasks shown in Figure 5. Compared to the sample efficiency of observing $M_{total} = 477000$ frames for solving 11 tasks by single-task agents, multitask agent has only observed $M_{total} = 29000 \times 11 = 319000$ frames when the last *social-media* task is solved as shown in Figure 5. Additionally, the plots indicate that multitask learning with simpler tasks is more efficient in using observed frames for hard tasks, achieving better $M_{task}$ than multitask learning with only those two tasks. These results indicate that our model enables positive transfers of learned behaviours between distinct tasks.

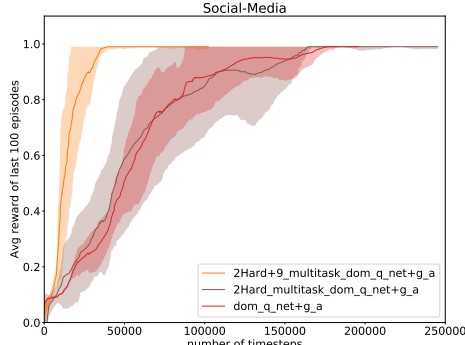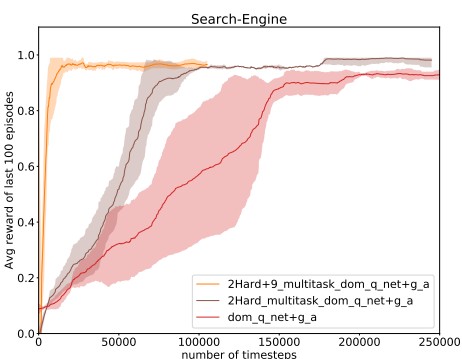

Figure 5: Comparisons in sample efficiency for 2 hard tasks, *social-media* (left) and *search-engine* (right), by multitask learning. *9_multitask* refers to the tasks discussed in Figure 4

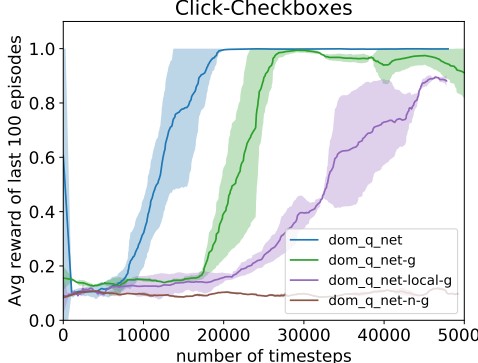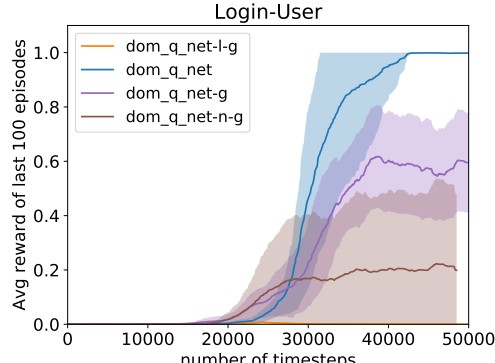

Figure 6: Ablation experiments for l=Local, n=Neighbor, g=Global modules. dom_q_net - g is the DOM-Q-NET without the global module. dom_q_net -l - g is the DOM-Q-NET with only neighbor module. dom_q_net-n-g is the DOM-Q-NET with only local module.

## 4.3 ABLATION STUDY ON THE DOM REPRESENTATION MODULES

We perform ablation experiments to justify the effectiveness of using each module for the $Q_{dom}$ stream. We compare the proposed model against three discounted versions that omit some modules for computing $Q_{dom}$: (a) $e_{dom} = e_{local}$, (b) $e_{dom} = e_{neighbor}$, (c) $e_{dom} = [e_{local}^T, e_{neighbor}^T]^T$.

Figure 6 shows the two tasks chosen, and the failure case for *click-checkboxes* shows that DOM selection without the neighbor module will simply not work because many DOMs have the same attributes, and thus have exactly the same representations despite the difference in the context. Liu et al. (2018) addressed this issue by hand-crafting the message passing. The faster convergence of DOM-Q-NET to the optimal behaviour indicates the limitation of neighbor module and how global and local module provide shortcuts to the high-level and low-level representations of the web page.

## 4.4 EFFECTIVENESS OF GOAL-ATTENTION

Most of the MiniWoB tasks have only one desired control policy such as "put a query word in the search, and find the matched link" where the word token for the query and the link have alignments with the DOMs. Hence, our model solves most of the tasks without feeding the goal representation to the network, with exceptions like *click-widget*. Appendix 6.7 shows comparisons of the model with different goal encoding methods including goal-attention. The effect of goal-attention is not obvious, as seen in some tasks. However, Figure 7 shows that the gain in sample efficiency from

using goal-attention is considerable in multitask learning settings, and this gain is much bigger than the gain in the single-task setting. This indicates that the agent successfully learns to pay attention to different parts of the DOM tree given different goal instructions when solving multiple tasks.

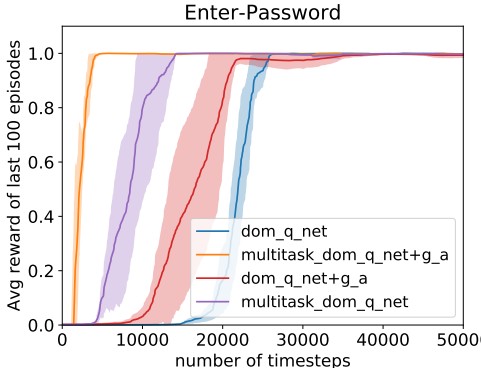 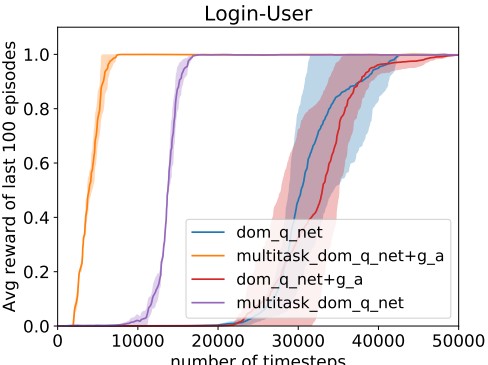

Figure 7: Effects of goal-attention for single and multi-task learning (g_a=goal attention)

## 5 DISCUSSION

We propose a new architecture for parameterizing factorized Q functions using goal-attention, local word embeddings, and a graph neural network. We contribute to the formulation of web navigation with this model. Without any demonstration, it solves relatively hard tasks with large action space, and transfers learned behaviours from multitask learning, which are two important factors for web navigation. For future work, we investigate exploration strategies for tasks like *email-inbox* where the environment does not have a simple instance of the task that the agent can use to generalize learned behaviours. Liu et al. (2018) demonstrated an interesting way to guide the exploration. Another work is to reduce the computational cost of evaluating the Q value for each DOM element. Finally, we intend on applying our methods to using search engines. Tasks like question answering could benefit from the ability of an agent to query search, navigate the results page and obtain relevant information for solving the desired goal. The ability to query and navigate search could also be used to bootstrap agents in realistic environments to obtain task-oriented knowledge and improve sample efficiency.

**Acknowledgement**: We acknowledge using the implementation of segment tree by Dopamine (Castro et al., 2018) for this project.

**Reproducibility**: Our code and demo are available at https://github.com/Sheng-J/DOM-Q-NET and https://www.youtube.com/channel/UCrGsYub9lKCYO8dlREC3dnQ respectively.

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

## 6 APPENDIX

### 6.1 HYPERPARAMETERS

Table 1: Hyperparameters for training with Rainbow DQN (4 components)

| Hyperparameter | Value |
|---|---|
| Optimization algorithm | Adam (Kingma & Ba, 2014) |
| Learning rate | 0.00015 |
| Batch Size | 128 |
| Discounted factor | 0.99 |
| DQN Target network update period | 200 online network updates |
| Number of update per frame | 1 |
| Number of exploration steps | 50 |
| N steps (multi-step) bootstrap | 8 |
| Noisy Nets $\sigma_0$ | 0.5 |
| Use DDQN | True |
| Easy Tasks: Number of steps for training | 5000 |
| Medium Tasks: Number of steps for training | 50000 |
| Hard Tasks: Number of steps for training | 2000000 |

Table 2: Hyperparameters for DOM-Q-NET

| Hyperparameter | Value |
|---|---|
| Vocabulary size: tag | 80 |
| Vocabulary size: text | 400 |
| Vocabulary size: class | 80 |
| Embedding dimension: tag | 16 |
| Embedding dimension: text | 32 |
| Embedding dimension: class | 16 |
| Dimension of Fully Connected(FC) layers | 128 |
| Number of FC layers for 3 factorized Q networks | 2 each |
| Hidden Layer Activation | ReLU |
| Number of steps for neural message passing | 3 (7 for social media task) |
| Max number of DOMS | 160 |
| Max number goal tokens | 18 |
| Out of Vocabulary Random vector generation | Choose-option, Click-Checkboxes |

Table 3: Hyperparameters for Replay Buffer

| Hyperparameter | Value |
|---|---|
| $\alpha$: prioritization exponent | 0.5 |
| $\beta$ for computing importance sampling weights | 0 |
| Single Task Buffer Size | 15000 |
| Multi Task Buffer Size | 100000 |

### 6.2 GOAL-ATTENTION OUTPUT MODEL

Figure 8 shows the readout phase of the graph neural network using goal-attention. The graph-level feature vector $h_{global}$ is computed by the weighted average of node-level representations processed with T steps of message passing, $\{h_1.....h_V\}$. The weights, $\{\alpha_1.....\alpha_V\}$, are computed with the goal vector as the query and node-level features as keys. For our model, we use a scaled dot product attention (Vaswani et al., 2017) with local embeddings as keys and neighbor embeddings as values, as illustrated in 3.3.

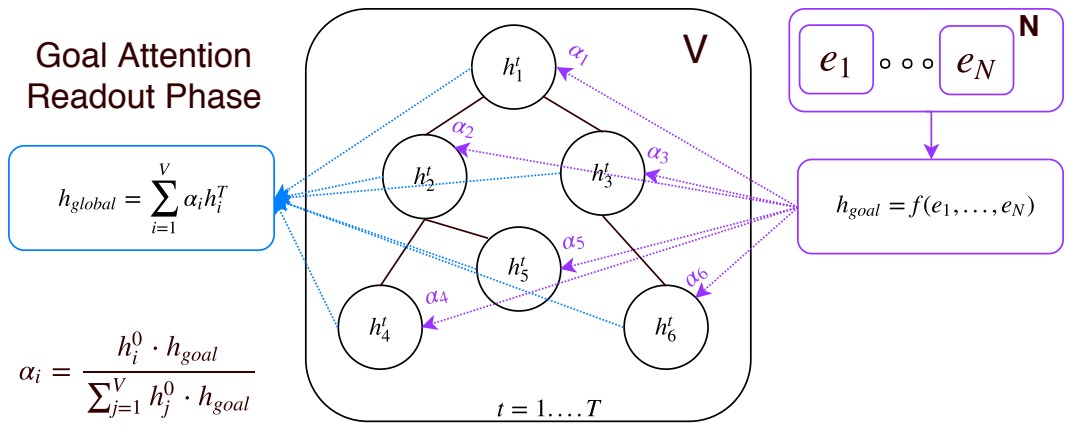

Figure 8: goal-attention

### 6.3 GOAL ENCODER

Three types of goal encoding module for global module are investigated.

1. Concatenating each node-level feature with the goal vector.
2. Goal-attention, as illustrated in 6.2
3. Using both *concatenation* and *attention*, as shown in Figure9

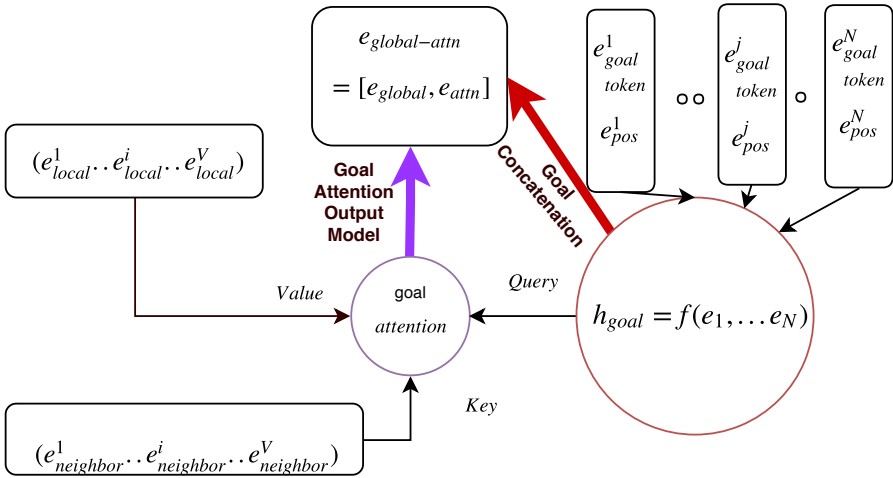

Figure 9: goal-encoder

Benchmark results for multitask and 23 tasks in Appendix 6.7 also compare the performances of using different goal encoding modules.

### 6.4 MINIWOB TASKS DIFFICULTIES DEFINITION

- **Easy Task**: Any task solvable under 5000 timesteps by single-task DOM-Q-NET {click-dialog, click-test, focus-text, focus-text-2, click-test-2, click-button, click-link, click-button-sequence, click-tab, click-tab-2, Navigate-tree}
- **Medium Task**: Any task solvable under 50000 timesteps by single-task DOM-Q-NET {enter-text, click-widget, click-option, click-checkboxes, enter-text-dynamic, enter-password, login-user, email-inbox-delete}

- **Hard Task**: Any task solvable under 200000 timesteps by single-task DOM-Q-NET, or any unsolvable tasks.
  {choose-date, search-engine, social-media, email-inbox}

## 6.5 MULTITASK LEARNING

---

**Algorithm 1** Multitask Learning with Shared Replay Buffer

---

1: **Given:**
  - an off-policy RL algorithm $\mathbb{A}$,          ▷ e.g. DQN, DDPG, NAF, SDQN
  - a set of environments for multiple tasks $\mathbb{K}$,     ▷ e.g. *click-checkboxes*, *social-media*
2: Initialize the shared replay buffer $R$
3: Initialize $\mathbb{A}$ by initializing the shared network parameters $\theta$
4: Initialize each environment and sample $s_0^{(k)}$
5: **for** i $= 1$ to $M$ **do**
6:     **for** each $k \in \mathbb{K}$ **do**
7:        Sample an action $a_t^{(k)}$ using behavioral policy from $\mathbb{A}$: $a_t^{(k)} \leftarrow \pi(s_t^{(k)})$
8:        Execute the action $a_t^{(k)} \rightarrow k$ and observe a reward $r_t^{(k)}$ a new state $s_{t+1}^{(k)}$
9:        Store the transition $(s_t^{(k)}, a_t^{(k)}, r_t^{(k)}, s_{t+1}^{(k)})$ in $R$
10:       Sample a minibatch $B$ from R, and perform one step optimization w.r.t $\theta$
11:       **if** episode for k terminated **then**
12:          reset k and sample $s_0^{(k)}$

---

## 6.6 EXPERIMENT PROTOCOL

We report the success rate of the 100 test episodes at the end of the training once the agent converges to its highest performance. The final success rate reported in Figure 3 is based on the average of success rate from 4 different random seeds/runs. In particular, we evaluate the RL agent after training it for a fixed number of frames, depending on the difficulty of the task, as illustrated in Appendix 6.4. As shown in table 4, the results presented in this paper is based on a total of 536 experiments using the set of hyperparameters in table 1, 2, 3.

Table 4: Experiment statistics

| | |
|---|---|
| Number of tasks | 23 |
| Number of tasks concurrently running for multitask | 9 |
| Number of goal encoding modules compared | 4 |
| $N_1 = (23 + 9) * 4 = 128$ | |
| Number of tasks for ablation study | 2 |
| Number of discounted models compared for ablation study | 3 |
| $N_2 = 2 * 3 = 6$ | |
| Number of experiments for computing the average of a result | 4 |
| Number of experiments for 11 multitask learning | 11 |
| $N_{total} = (128 + 6 + 11) * 4 = 580$ | |

## 6.7 BENCHMARK RESULTS

We present the learning curves of both single-task and multitask agents. We also provide the learning curves of the model with different goal-encoding modules 6.3. X-axis represents the number of timesteps, and Y-axis represents the moving average of last 100 rewards. For medium and hard tasks, we also show the fraction of transitions with positive/non-zero rewards in the replay buffer and the number of unique positive transitions sampled throughout the training. This is to demonstrate the sparsity of the rewards for each task, and investigate whether the failure comes from *exploration*.
Note that we are using multi-step bootstrap (Sutton, 1988), so some transitions that do not directly lead to the rewards are still considered "positive" here.

### 6.7.1 MULTITASK (9 TASKS) RESULTS

The following plots show the learning curves for the 9 tasks used in multitask learning.

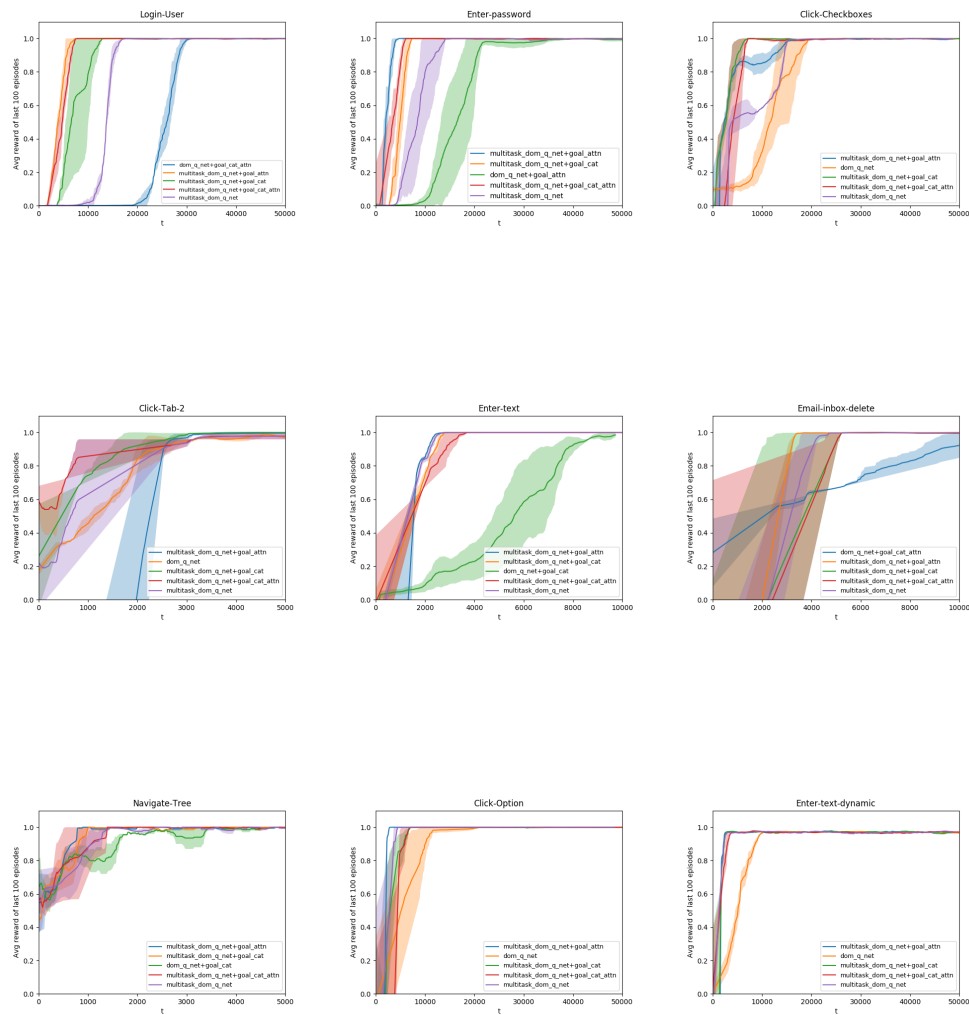

### 6.7.2 SOME EASY AND MEDIUM TASKS

We omit the plots for very simple tasks requiring less than 1000 training steps.

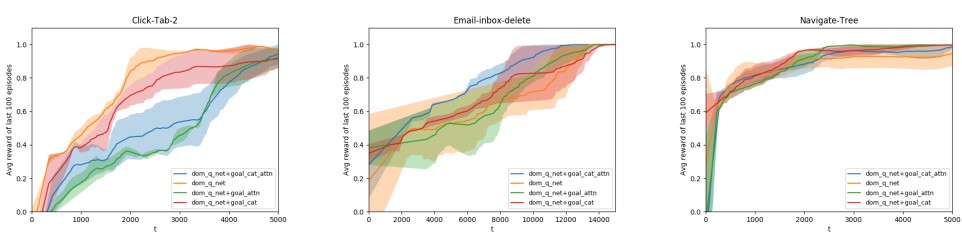

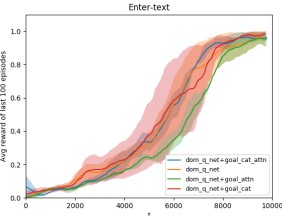

### 6.7.3 MEDIUM TASKS WITH REPLAY BUFFER INFORMATION

The plots on the left show the moving average of the rewards for last 100 episodes.

The plots on the center show the fraction of positive transitions in replay buffer.

The plots on the right show the unique number of positive transitions for each training batch.

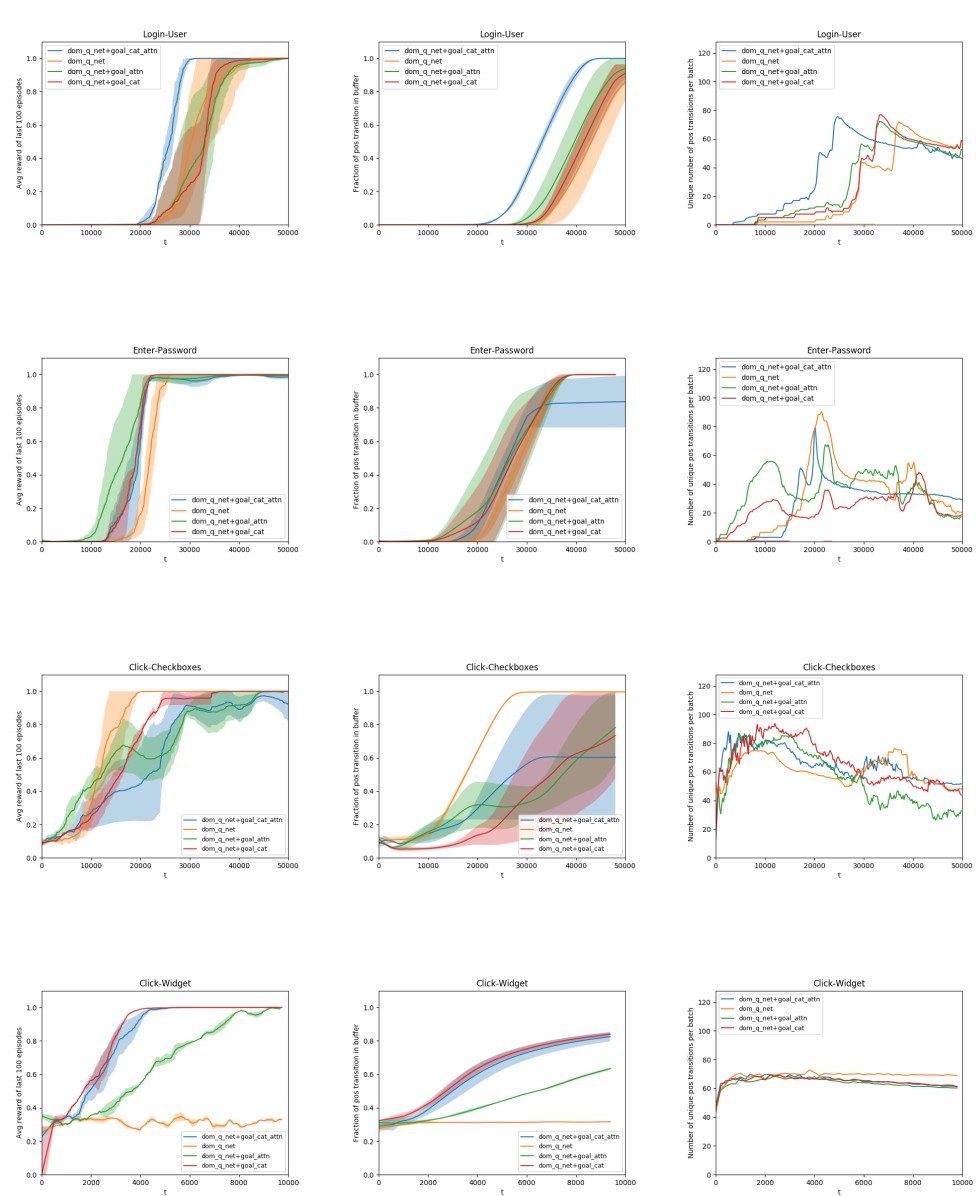

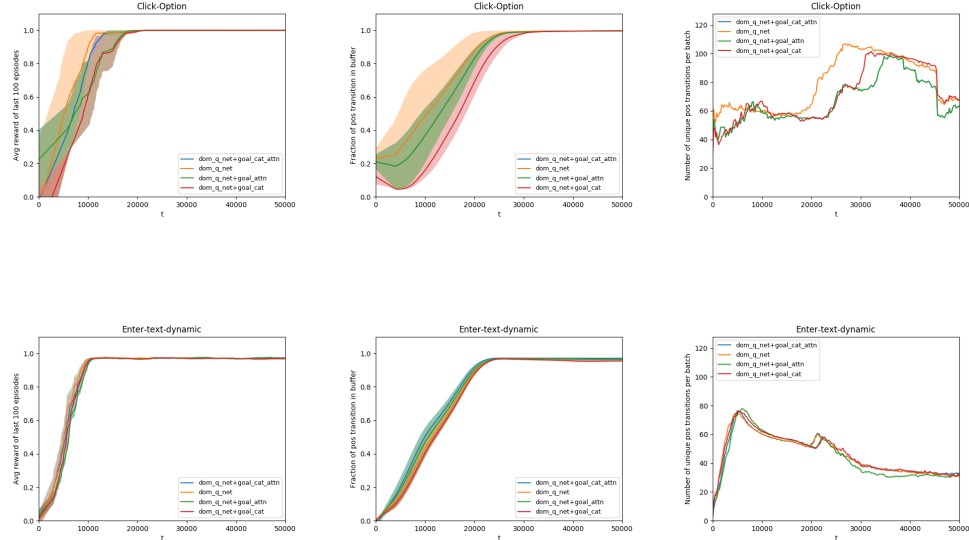

### 6.7.4 HARD TASKS

The plots on the left show the moving average of the rewards for last 100 episodes.

The plots on the center show the fraction of positive transitions in replay buffer.

The plots on the right show the unique number of positive transitions for each training batch.

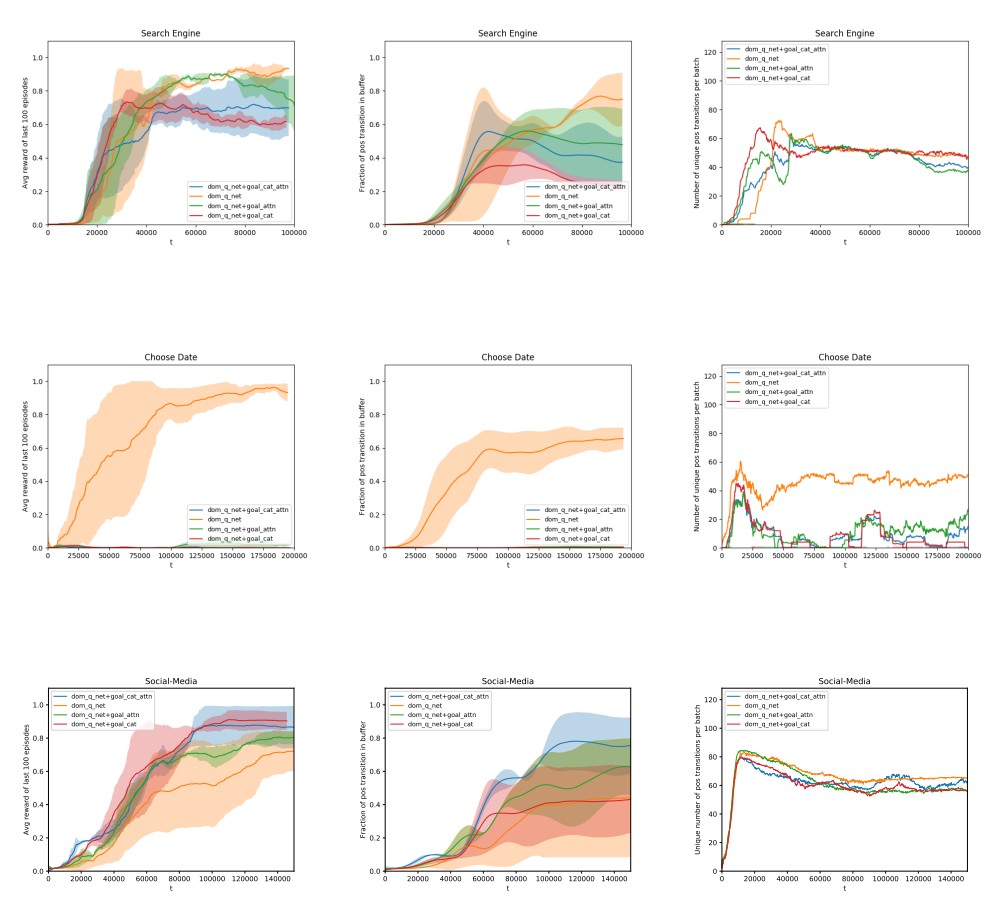

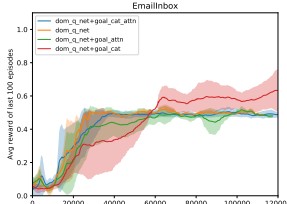 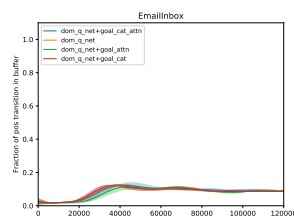 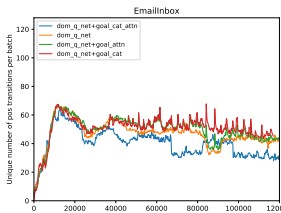

