# OpenReview forum: "DOM-Q-NET:  Grounded RL on Structured Language"
_ICLR.cc/2019/Conference_

### Official Review · AnonReviewer2 · 2018-11-02

**Rating:** 6
**Confidence:** 3

**Review:**

DOM-Q-NET: GROUNDED RL ON STRUCTURED LANGUAGE

This paper presents a somewhat novel graph-based Q-learning method for web navigation benchmark tasks. The authors show that multi-task learning helps in this case and their method is able to learn without BC as previous works have needed. While this work is interesting and to my knowledge somewhat novel. I concerns with one aspect of the evaluation. In some part it was stated that they show the highest success rate for testing on 100 episodes, if this is indeed the maximum success rate, it is unclear if these results are misleading or not. It is possible that there was a lucky seed in those 100 episodes leading to a higher max that is not representative of the algorithm performance. Also, please have the submission proof-read for English style and grammar issues. There are many minor mistakes, some of which are pointed out below. I am rating marginally below due mainly to the potentially misleading results from the comment on using the highest success rate to report results and to a minor extent due to the novelty aspect (though this is an interesting application).


Comments:

- “Evaluation metric: we plot the moving average of the reward for last 100 episodes, and report the highest success rate for testing on 100 episodes.” —> This is unclear, do you mean you only displayed the maximum success rate out of all 100 episodes? So if the success rates are [0, 100, 0, 0, 0], Figure 2 shows 100% success? If so, this is somewhat misleading and a better metric may have been the average success rate with confidence intervals. Otherwise you may have just gotten a lucky random seed potentially.
- I would’ve liked to see if this is the only method which benefits from multitask learning or do DOMNETs also benefit. This however, is just a nice to have.
- I appreciate the inclusion of hyper parameters and commitment to releasing the code in an effort to promote reproducibility! Great job there.
- I really like the idea of using graph networks with RL, though I’m not sure if it’s novel to this work. Interesting line of work!
- While this is an interesting application, I’m not sure about the novelty. I suggest spending a bit more time discussing how this work contrasts with methods like Wang et al., or others cited here.

Typos:

“MiniWoB(Shi et al., 2017) benchmark tasks. “ —> missing space between citation
“Q network architecture with graph neural network” —> with a graph neural network
"MiniWoB(Shi et al., 2017)” —> MiniWoB (Shi et al., 2017) (missing space)
“achieved the state” —> achieved state of the art
“2016; Wang et al., 2018)as main” —> missing space
“series of attentions between DOM elements and goal” —> series of attention (modules?) between the DOM elements and the goal (?)
“constrained action set” —> constrained action sets
“In appendix, we define our criteria for difficulties of different tasks.” —> In the appendix

---

> ### Author Response · Authors · 2018-11-23
> **Response to AnonReviewer2 (PART1)**
>
> Dear reviewer,
> Thank you for taking the time to review our paper.
> We appreciate the valuable comments that improve the readability and the clarity of the paper and we will incorporate all the changes and fix the typos in our latest revision.
>
> [Concern1] “Unclear evaluation metric”
> [Reply]  Our experimental protocol follows the previous works on the same environment[1, 4]. We report the success rate of the 100 test episodes at the end of the training once the agent has converged to its highest performance on the training episodes. The final success rates reported in Figure 2 of the original submission were averaged across 4 different random seeds/runs. We apologize for the poor wording that has been corrected in our latest revision.
>
> In detail, what we did was to evaluate the RL agent after training for a fixed number of frames depending on the difficulty of the task.  In the original paper, we mentioned in the appendix that we used three different number of frames {5000, 50000, 200000} for training based on the difficulty {easy, medium, hard} of the 23 tasks in MiniWOB.  In the initial experiments, we observed that some tasks were solved with far less number of frames than others due to varying difficulties, so we categorized 23 tasks in three difficulty groups to shorten the experiment time for simpler tasks. This alleviated unnecessary computational cost for a large number of experiments. The results and the plots we presented in the paper, are based on the following number of experiments.
> Number of experiments = (23(number of tasks) + 9(number of tasks concurrently running in multitask) ) * 4(types of goal encoding) * 4 (minimum num of runs for average)  + 2(tasks for ablation study) * 3(discounted model) * 4(minimum num of runs for average)= 536 experiments for one set of hyperparameters.
> For further details on our experiment protocols, please check the updated "evaluation metric" in Sec4.1 and Appendix6.5
> So our success rate reported in Figure 2 (original paper) is based on the average success rate of  4 runs.
>
>
> [Concern2] “Novelty issue”, "Lack of comparisons with previous works on GNN+RL"
> [Reply]  To our knowledge, this is the first work that applies graph neural networks (GNNs) to represent the HTML structure in standard web pages. This leads to our novel deep Q-network architecture that incorporates both the goal attention mechanism and the GNN representation to learn the state-action value function for Q learning. We appreciate the reviewer to point out the similarity and lack of comparisons with other GNN+RL models, e.g. Nervenet[2].  Unfortunately, previous works on GNN+RL are not directly applicable to our web navigation problem. Please see the following paragraph that has been added to the latest revision for a detailed explanation.
>
> - Our main contribution is to propose a new architecture for parameterizing factorized Q functions using goal attention, local word embeddings and graph neural network(GNN).  We also contributed to the formulation of web navigation with this architecture.   GNN is one of the components, and we investigated in the ablation study that some tasks need GNN for neural message passing [3] and some tasks do not necessarily need it though the sample efficiency is better with GNN.  We also showed how proposed goal attention can be used with GNN for even better sample efficiency when multitasking.  Computing the output model of GNN with goal attention is unique in our goal-oriented RL setting with graph state/action representations.   Previously proposed Graph Attention Networks [5] uses attention in neural-message passing phase, and is experimented in non-RL settings.  In general, GNN is not actively used in RL settings as seen in this comprehensive survey paper of GNN [6], and the previous papers [2, 7] use GNN for mimicking the physical bodies of different robots.  We would like to mention some differences when using GNN for representing web pages.

---

> > ### Comment · AnonReviewer2 · 2018-11-24
> > **Appreciate revision and reply**
> >
> > Thank you for your reply. I think this clears up some concerns I had, and I appreciated the added detail in the paper/appendix. While I still have minor concerns about novelty, I believe the new text helps clear up most of this. I've updated my original rating as such.

---

> ### Author Response · Authors · 2018-11-23
> **Response to AnonReviewer2 (PART2)**
>
> There are four key differences that makes NerveNet [2] not applicable to our web navigation task:
> 1) In Nervenet [2], the entire policy is parametrized by GNN.  In DOM-Q-NET, GNN alone cannot solve some tasks for 100% success rate. We conducted a careful ablation study on the different modules of our proposed Q-network architecture in Figure 4(original paper)/ Figure 5(updated revision).  We also conducted the experiments with and without goal attention shown in Figure 2(original paper).   The Login-user task and the Social Media task, for example, cannot be solved using GNN component alone for Q_dom network.
>
> 2) In Nervenet [2] for locomotion, action is goal independent.  In DOM-Q-NET for web navigation, action is goal dependent, which is why goal attention is proposed.
>
> 3) In Nervenet [2], graph structure is static across different timesteps within an episode. In DOM-Q-NET, graph structure is dynamic even within an episode because the web page can change after an action.
>
> 4) Nervenet [2] learns its controller model using on-policy policy gradient with dense reward from the locomotion control tasks. However, the web navigation tasks are spare reward problems with only 0/1 reward at the end of the episode, DOM-Q-NET uses off-policy Q-learning with replay buffer that is often more efficient against reward sparsity.
>
> We will make a further clarification in our background section to make this distinction clear.
>
>
> [Concern3] “do DOMNET’s also benefit from multitask learning”
> [Reply]  Thank you for bringing this interesting question. DOMNETs [1] take in “key-value goal” as an input in addition to natural language goal.  For example, the task with the goal “click A” and the task with the goal “click A and press B” will have different size of inputs; 1 for [“A”] and 2 for [“A”, “B”].  The embeddings for those key-value inputs are directly fed into the network without being aggregated. So the dimension of the weight matrices of the DOMNET is task-dependent.   It is mentioned in their paper that structured input is needed for workflow policy as a result of using the formal language for constraining actions.  Therefore, it is not trivial to extend a single DOMNET model with shared weight matrices to multitask learning.
>
> In addition, we have investigated whether the model benefits more from multitasking with various methods of incorporating goal information. “Goal-attention” leads to better sample efficiency without any increase in the network size.  This is shown in “effectiveness of goal attention”.   DOMNET, however, does not have a flexible attention module with GNN to incorporate different goals that may hinder its performance in multitask learning.
>
> Please let us know if you have any more questions or if there is anything else we can clarify to make you reconsider your rating.
>
> [1] Evan Zheran Liu, Kelvin Guu, Panupong Pasupat, Tianlin Shi, and Percy Liang. Reinforcement learning on web interfaces using workflow-guided exploration. In ICLR, 2018.
> [2] Tingwu Wang, Renjie Liao, Jimmy Ba, and Sanja Fidler. Nervenet: Learning structured policy with graph neural networks. In ICLR, 2018.
> [3] Justin Gilmer, Samuel S Schoenholz, Patrick F Riley, Oriol Vinyals, and George E Dahl. Neural message passing for quantum chemistry. 2017
> [4] Tianlin Shi, Andrej Karpathy, Linxi Fan, Jonathan Hernandez, and Percy Liang. World of bits: An open-domain platform for web-based agents. In ICML, 2017.
> [5] Petar Velickovic, Guillem Cucurull, Arantxa Casanova, Adriana Romero, Pietro Lio, and Yoshua Bengio. Graph attention networks. In ICLR, 2018
> [6] Peter W Battaglia, Jessica B Hamrick, Victor Bapst, Alvaro Sanchez-Gonzalez, Vinicius Zambaldi,
> Mateusz Malinowski, Andrea Tacchetti, David Raposo, Adam Santoro, Ryan Faulkner, et al.  Relational inductive biases, deep learning, and graph networks.
> [7] Hamrick, J., Allen, K., Bapst, V., Zhu, T., McKee, K., Tenenbaum, J., and Battaglia, P. (2018).  Relational inductive bias for physical construction in humans and machines. In CogSci, 2018

---

### Official Review · AnonReviewer3 · 2018-11-04
**Interesting paper**

**Rating:** 7
**Confidence:** 1

**Review:**

The authors propose a novel architecture for RL-based web navigation to address both of these problems, DOM-Q-NET, which utilizes a graph neural network to represent tree-structured HTML along with a shared state space across multiple tasks. It is believed more flexible to be probed on WorldOfBits environments. Significant improvements are shown by experiment.

---

> ### Author Response · Authors · 2018-11-23
> **Response to AnonReviewer3**
>
> Dear reviewer,
> Thank you for the positive feedback and taking the time to review our paper.   In order to pursue further reproducibility, we clarified our experiment protocols in sec4.1 and the appendix 6.5 for further details.  We have also added background sec2.5 and the appendix 6.7 for further details on a task solved by semantic parsing and how it is different from MiniWoB.  In addition, a demo of a successful trajectory, figure2 in the revision, is added to further demonstrate our problem setup and the instances for the tuple of actions.

---

### Official Review · AnonReviewer4 · 2018-11-10
**Moderately interesting result, but not aware at all of a huge related literature**

**Rating:** 7
**Confidence:** 3

**Review:**

Caveat: I am an emergency reviewer filling in for someone that fell through on their commitment to review for ICLR.  The framing of this paper is quite outside my typical area, so I am not super familiar with the related work here, nor do I have time to get familiar with it for this last-minute review.

This paper presents a new model for deep reinforcement learning on web pages, where the system is given a goal (stated in text) and is supposed to interact with the web page (through clicking and entering text) in order to achieve that goal.  The supervision is a positive reward when the sequence of actions taken matches the goal.  The novel model presented in this paper is a modular Q function that incorporates graph embeddings of the web page's DOM, as well as similarity scores between elements in the DOM with words in the goal.

Just judging the presentation of the paper, it looks sound.  The methods seem reasonable (very similar to methods that are known to work well on related problems; more on that below), and the experiments look to be well done.  The paper is reasonably well written.  I don't know the RL community well enough to know how impactful this particular piece of work would be there - it's a new model architecture, basically, that gives improved performance.  I'd probably give a similar paper in my area a 3.5-4 out of 5 for an ACL conference.  The one major drawback I see in this paper is that it is _so_ similar to work on semantic parsing, but doesn't realize it.

I am not a "reinforcement learning" researcher, though I am a "semantic parsing" researcher.  The problem statement in this paper reads to me exactly like a semantic parsing problem: map a piece of text to a statement in some formal language.  In this case, the "statement" is a sequence of actions on the DOM of a web page.  The web page is possibly unseen at test time (the particulars of the data setup weren't totally clear to me), so the model has to be able to handle linking words in the sentence to pieces of the DOM in a way that doesn't rely on having seen those DOM elements during training.  This setup seems almost identical to the WikiTableQuestions dataset (Pasupat and Liang 2015), which has seen several RL-inspired works recently (e.g., https://arxiv.org/abs/1807.02322).  The way that the authors propose to use attention scores in the "global module" is _very_ similar to the linking mechanism proposed by Krishnamurthy, Dasigi and Gardner (EMNLP 2017) for WikiTableQuestions, and the way that the "word-token selection" only allows words in the goal sentence is very reminiscent of Chen Liang's language for parsing questions in WikiTableQuestions, which has similar restrictions for similar reasons.

I think the main difference between what we call "weakly-supervised semantic parsing" and what you call "deep reinforcement learning" is that semantic parsing leverages the fact that we know the language we're parsing into, so we don't need to use model-free RL methods like Q-learning.  We know the model, so we can be much smarter about learning.  Again, I'm not super familiar with the tasks you're looking at here, but I'm pretty sure there are much better _supervised_ learning techniques that you could apply to these problems.

All of this is to say that the methods proposed here look _very_ similar to methods that have been studied for quite a while in the semantic parsing literature (I gave only recent references above, but the basic problems go back decades; e.g., http://aclweb.org/anthology/P09-1010, or http://www.cs.utexas.edu/~ml/papers/senior-aaai-2008.pdf).  Yet this paper only cites recent deep RL papers.  I think the authors would benefit greatly from familiarizing themselves with this literature.  I think the semantic parsing community would also benefit from this, as there are surely ideas in the deep RL community that we could benefit from, too.  But the two communities don't really talk to each other much, it seems, even though in some cases we are working on _very_ similar problems.

So, to summarize: the paper seems reasonable enough.  I'm guessing that the RL community would find it at least moderately interesting, and it appears well written and well executed.  My one concern is that it's totally oblivious to the fact that it's sitting right next to a well-established literature that could probably teach it a thing or two about mapping language to actions.


--------------

After seeing the authors engage at least a little with the related semantic parsing literature, I've increased my score to a 7.

---

> ### Author Response · Authors · 2018-11-23
> **Response to AnonReviewer4**
>
> Dear reviewer,
> We thank the reviewer for the valuable comments in pointing out the connection between our work and semantic parsing.  We have updated the background section to discuss the semantic parsing methods to solve the problem like Question Answering from manipulating the data on HTML tables.
>
> We would like to emphasize that the main focus of this work is to train an end-to-end RL agent to directly interact with any standard web browser through mouse clicking and typing. Our direct approach avoids the challenge of designing the base-level primitive logical forms in semantic parsing for web navigation.
>
> The problem setup of mapping language to click/type action was introduced in the MiniWoB[7], which is a set of standard benchmark environments for web agents. The authors of [7], in fact, found that reinforcement learning approaches often significantly outperform supervised learning on these benchmark tasks. We have added a new illustration in Figure 2 to further clarify our problem setup.
>
> [Concern] “This setup seems almost identical to the WikiTableQuestions dataset (Pasupat and Liang 2015), which has seen several RL-inspired works recently”
> [Reply] We appreciate pointing out the relevant works, and we explained the problem setup of this dataset [5] as well as the difference between this task and web navigation in the background sec2.5 and the appendix 6.7 for further details.  In short, WikiTableQuestions[5] provides structured HTML tables that only contain the text attributes from the original HTML page. To execute the parsed logical form, an executor is provided. However, MiniWoB[7] environment has a set of more diverse web pages beyond tables. The agent needs to understand raw HTMLs that contain text fields, buttons and checkboxes. Each MiniWoB task is given by natural language goal instructions on different web pages. This RL environment only accepts basic actions like “click DOM indexed i”, “Type a string on DOM indexed i”.   So we took a direct approach that allows our agent to control clicking and typing. which avoids the challenge of designing rich primitives and formal language for web navigation.
>
> [Concern] “The problem statement in this paper reads to me exactly like a semantic parsing problem”
> [Reply] Our main goal is to train an agent end-to-end to directly click the DOMs and type strings on the standard browser.  We are converting natural language to a sequence of clicking and typing actions that a browser can execute.
> “Map a piece of text to a statement in some formal language” - This is what we hope to avoid because the standard browser only accepts “which DOM to type/click”, “what to type” as valid actions, so we cannot have a complex logical form as an output of the model.  For web navigation, it is not trivial to design formal language and the primitives of the formal language.   However, we noted in the section 2.6 that Liu et al [6] defined their minimalistic formal language to constrain the exploration, but they still use RL to perform the same set of actions as ours to a standard browser.  The focus of our work is to learn an end-to-end RL agent that can act directly in a web browser. Empirically, we found our end-to-end agent matches and outperforms (for some tasks) the models augmented with formal language studied in Liu et al[6].
>
> [1] Jayant Krishnamurthy, Pradeep Dasigi, and Matt Gardner. Neural semantic parsing with type constraints for semi-structured tables. In Proceedings of the 2017 Conference on Empirical Methods in Natural Language Processing, pp. 1516–1526. Association for Computational Linguistics, 2017. doi: 10.18653/v1/D17-1160. URL http://aclweb.org/anthology/D17-1160.
>
> [2] Satchuthananthavale RK Branavan, Harr Chen, Luke S Zettlemoyer, and Regina Barzilay. Reinforcement learning for mapping instructions to actions. In Proceedings of the Joint Conference of the 47th Annual Meeting of the ACL and the 4th International Joint Conference on Natural Language Processing of the AFNLP: Volume 1-Volume 1, pp. 82–90. Association for Computational Linguistics, 2009.
>
> [3] Chen Liang, Mohammad Norouzi, Jonathan Berant, Quoc Le, and Ni Lao. Memory augmented policy optimization for program synthesis with generalization. arXiv preprint arXiv:1807.02322, 2018
>
> [4] Raymond J Mooney. Learning to connect language and perception. In AAAI, pp. 1598–1601, 2008.
>
> [5] Panupong Pasupat and Percy Liang. Compositional semantic parsing on semi-structured tables. arXiv preprint arXiv:1508.00305, 2015.
>
> [6] Evan Zheran Liu, Kelvin Guu, Panupong Pasupat, Tianlin Shi, and Percy Liang. Reinforcement learning on web interfaces using workflow-guided exploration. In ICLR, 2018.
> [7] Tianlin Shi, Andrej Karpathy, Linxi Fan, Jonathan Hernandez, and Percy Liang. World of bits: An open-domain platform for web-based agents. In ICML, 2017.

---

> > ### Comment · AnonReviewer4 · 2018-11-25
> > **Reply**
> >
> > I appreciate the attempt to put in a section of related work on semantic parsing, but my goal was not to get you just to cite work on WikiTableQuestions (WTQ), but to see the similarities between what you are doing and what semantic parsing does.  It doesn't appear that you've actually accepted those similarities, and the section you added (section 2.5) feels out of place in the paper, because it doesn't discuss how that work is related to the current paper.
> >
> > To be more concrete: the reason I brought up WTQ was not because it operated on HTML tables.  It was because it mapped language to actions in unseen contexts, and methods developed for WTQ are _very_ similar to the methods you developed in your paper.  You say here that you "avoid the challenge of designing the base-level primitive logical forms" - actually, the action space that you have _is_ the "base-level primitive logical form" language in this context.  You have the same basic task description, just with a language of clicks and text boxes instead of counts and argmaxes.
> >
> > If anything, the updates I see in the paper make me less inclined to recommend an "accept", because they look like they were an attempt to appease a grumpy reviewer with additional irrelevant citations.  The issue is not navigating HTML vs. answering questions, and section 6.6.7 is entirely unnecessary - I know the differences between the two, as would anyone who is familiar with them.  That section adds nothing to the paper.  The issue is higher level than that.  The broader area that you're operating in is "mapping language to actions", with navigating web pages just one particular instance of this very general, well-studied problem.  In order to contribute to this literature, you need to understand how your work relates to the literature.  It turns out that much of your contributions can already be found there, if you know how to look, and you aren't situating your work in the context of what others have already done.  This is a problem.

---

> > > ### Author Response · Authors · 2018-11-27
> > > **Response [Part2]**
> > >
> > > 3)  Model architecture difference
> > > Knowledge graph embedding used in [1] is a bag-of-word-vectors model that aggregates the information by simply summing over all the word vector in the neighborhood of an entity with a human engineered rule.  This is similar to our model using only the local and global module. We found that the current state-of-the-art method[2] for WQT adopts the Neural Symbolic Machines Framework[6] that is a variant of the seq2seq encoder-decoder model.
> > >
> > > Our proposed model incorporates graph neural networks that learn a flexible nonlinear message passing function over the entire input graph converted from raw HTML. We show that having the message passing phase, e_nieghbor module is crucial in solving many of the MiniWoB tasks as shown in the ablation study Sec 4.3.
> > > This is simply because many ‘entities’ in web like ‘checkboxes’ have exactly the same embeddings and multi-step message passing is required to propagate the text values, Figure 1.  As far as we can tell, there are no prior works on WTQ using graph neural networks.
> > >
> > > 4) Training objective function: marginal log-likelihood vs Q learning
> > > In semantic parsing, [1] first generates a set of logical forms that execute to the correct answer, and enumerates those correct forms for each example and use the sum of log-likelihood of generating those correct forms as objective.
> > > In our setup,  we aim to solve the problem of dynamic programming directly with Q learning to minimize the TD error.
> > >
> > > In summary, there are some similarities between WTQ tasks and a reinforcement learning approach to web navigation in terms of using word embeddings and deep neural models. However, our proposed problem formulation and model architecture is significantly different than the methods that were previously developed on WTQ.
> > >
> > >
> > > [1] Jayant Krishnamurthy, Pradeep Dasigi, and Matt Gardner. Neural semantic parsing with type constraints for semi-structured tables. In Proceedings of the 2017 Conference on Empirical Methods in Natural Language Processing, pp. 1516–1526. Association for Computational Linguistics, 2017. doi: 10.18653/v1/D17-1160. URL http://aclweb.org/anthology/D17-1160.
> > >
> > >
> > > [2] Chen Liang, Mohammad Norouzi, Jonathan Berant, Quoc Le, and Ni Lao. Memory augmented policy optimization for program synthesis with generalization. arXiv preprint arXiv:1807.02322, 2018
> > >
> > > [3] Panupong Pasupat and Percy Liang. Compositional semantic parsing on semi-structured tables. arXiv preprint arXiv:1508.00305, 2015.
> > >
> > > [4] Evan Zheran Liu, Kelvin Guu, Panupong Pasupat, Tianlin Shi, and Percy Liang. Reinforcement learning on web interfaces using workflow-guided exploration. In ICLR, 2018.
> > >
> > > [5]  Y. Wang, J. Berant, and P. Liang, “Building a Semantic Parser Overnight,” Proc. 53rd Annu. Meet. Assoc. Comput. Linguist. 7th Int. Jt. Conf. Nat. Lang. Process. (Volume 1 Long Pap., pp. 1332–1342, 2015.
> > >
> > >
> > > [6] Chen Liang, Jonathan Berant, Quoc Le, Kenneth D Forbus, and Ni Lao. Neural symbolic machines: Learning semantic parsers on freebase with weak supervision.
> > > arXiv preprint arXiv:1611.00020, 2016.
> > >
> > > [7] Tianlin Shi, Andrej Karpathy, Linxi Fan, Jonathan Hernandez, and Percy Liang. World of bits: An open-domain platform for web-based agents. In ICML, 2017.

---

> > > ### Author Response · Authors · 2018-11-27
> > > **Response [Part1]**
> > >
> > > We appreciate your prompt reply.  We are sorry that our update in the background section for semantic parsing did not show relevance to our work, and we decide to temporarily revert the part of our revision and remove this section and its appendix 6.6.7.  As “reinforcement learning” researchers, we tried our best to familiarize ourselves to semantic parsing and make relevant connections to our work within this period.  Since previous works for our problem setup [4, 7] also did not mention connections with semantic parsing, this concern is novel, and we hope to explore more on the connections in the later revision of the paper. Based on our understandings, we still believe there are some major differences in what we are doing and what semantic parsing is doing.
> > >
> > > [Concern]  "The methods developed for WTQ are __very__ similar to the methods you developed in your paper"
> > > [Reply] We respectfully disagree with this statement. The problem formulations and methods developed for WTQ and the web navigation tasks in our paper are significantly different.  Here, we provide four major differences:
> > >
> > > 1) Act in partially observable environment vs fully observed knowledge graph
> > > In WTQ, we aim to learn an agent that can map language to action given the FULL access to the knowledge graph. As the agent can query the knowledge graph freely in the WQT tasks, there is no need for exploring the knowledge graph to gather previously unseen information. As a result, actions like argmax and count can be performed easily on the given knowledge graph.  (E.g. R[λx[Year.Date.x]].argmax(Country.Greece,Index), operation argmax can be performed only when the full knowledge graph is observed )
> > >
> > > On contrary, our problem formulation assumes the agent lives in a partially observed world. At each timestep, the input to the agent is ONE of the many web pages it can 'navigate' to on a website. Most information contained in the website is NOT present on the current webpage. To solve a language query, the agent has to search for relevant contents by visiting other web pages. For example, in the MiniWoB social media task, the episode starts with page 1 out of 10 twitter pages. To answer the query, our agent needs to use the Twitter interface to flip through the pages. The input web page changes after each click/type action. In other words, our problem formulation is to map language to __a sequence of__ actions. After each action, the input web pages or knowledge graph changes. Therefore, we formulated the web navigation problem as a sequential decision-making process due to the partially observable environment. This differs from the single decision-making problems formulated by all the methods developed for WTQ using semantic parsing.
> > >
> > > Solving with partially observable knowledge is essential to web navigation as it will require significant works in crawling all the web pages to provide the full knowledge.
> > >
> > >
> > > 2) base level primitive of click/type vs argmax/sort/...etc
> > > To solve WTQ tasks, all the approaches assume a domain-specific grammar(e.g. Lisp domain specific language) [1 2 3 5] and specific domain knowledge. In [1], the authors design grammars that contain type-constraints to associate entities to their correct types in the generated logical forms, e.g., disallow the action "country = 120" with the scope that pre-defines typed variable bindings.   [2] also shows the need for having code assistance by eliminating syntactically or semantically invalid choices.   A drawback of these existing methods developed for WTQ is that they still require a great deal of human supervision. The role of hand-crafted grammars is crucial in WTQ yet also limits its general applicability to many different domains. In [5], the authors build semantic parsers for 7 domains, hand-engineered a separate grammar for each domain.
> > >
> > > Designing high-level domain specific grammar may lead to faster learning for some problem domains but will restrict/bias the agent by human-engineered rules.  Previous work with DOMNET[4] did design minimalistic formal language to restrict its action space. we argue learning to act using the low-level primitives, e.g. ‘click’ and ‘type’, shared among many tasks allows the agent to transfer learned ‘high-level’ behaviors to unseen domains/websites.  This is shown in the improvements in sample efficiency when we train a single model to solve many of the MiniWoB tasks through shared low-level primitives ‘click’, ‘type’, see Sec 4.2 on multi-task learning.

---

> > > > ### Comment · AnonReviewer4 · 2018-11-27
> > > > **Thanks, nice summary**
> > > >
> > > > Thank you, this was much more of what I was looking for in my initial review.  I agree with you that there are substantial differences (e.g., I hadn't considered the partially observable environment, thanks for pointing that out), but there are also some very close similarities, and reinforcement learning methods are starting to intersect more with semantic parsing (in addition to the citations you already have, here's another good one: https://arxiv.org/abs/1704.07926).  Both of our fields would benefit from more discourse, and all I was hoping for was for you to engage a little bit with this literature.  You've done that in what you just wrote.  I don't think there's any need to mention or cite things that aren't relevant to your paper just because I mentioned it, but I think some distilled version of what you have here with the most relevant bits would be a nice section to add to the paper.
> > > >
> > > > I've increased my score to a 7.

---

> > > > > ### Comment · AnonReviewer4 · 2018-11-27
> > > > > **One more brief note**
> > > > >
> > > > > A minor point, just in case it's helpful (apologies if you already know this): one of the main goals of writing a related work section like this is to get the authors of that related work interested in what you're doing, to convince them to try your methods.  So, e.g., "we do something similar to the knowledge graph embedding of Krishnamurthy et al, but we do it better for these reasons" might make those authors look at and possibly use your work, where they otherwise might never know about it.  I'd look at writing a semantic parsing related work section as an opportunity to appeal to those people who are working on similar problems and expand the influence of your paper (especially as the main contributions here seem to be modeling contributions, and those are the ones most easily transferable between semantic parsing and RL).

---

### Meta-Review · Area_Chair1 · 2018-12-18
**Novel application of RL, sound results**

**Confidence:** 3
**Recommendation:** Accept (Poster)

**Metareview:**

This paper considers the task of web navigation, i.e. given a goal expressed in natural language, the task is to navigate webs by filling up fields and clicking links. The proposed model uses reinforcement learning, introducing a novel extension where the graph embedding of the pages is incorporated into the Q-function. The results are sound, and the paper is overall well-written.

The reviewers and AC note the following potential weaknesses. The primary concern that was raised was the novelty. Since the task could potentially be framed as semantic parsing, reviewer 4 mentioned there may be readily available approaches for baselines that the authors did not consider. The comparison to semantic parsing required a more detailed discussion, pointing not only the differences but also the similarities, that would encourage the two communities to explore novel approaches to their tasks. Further, reviewer 2 was concerned about the limited novelty, given the extensive work that combines GNN and RL, such as NerveNet.

The authors provided comments and a revision to address these issues. They described why it is not trivial to formulate their setup as a semantic parsing problem, partly due to the fact that the environment is partially observable.
Similarly, the authors described the differences between the proposed approach and methods like NerveNet, such as the use of a dynamic graph and off-policy RL, making the latter not a viable baseline for the task. These changes addressed most of the concerns raised by the reviewers.

The reviewers agreed that this paper should be accepted.